# Sleep and Economic Status Are Linked to Daily Life Stress in African-Born Blacks Living in America

**DOI:** 10.3390/ijerph19052562

**Published:** 2022-02-23

**Authors:** Zoe C. Waldman, Blayne R. Schenk, Marie Grace Duhuze Karera, Arielle C. Patterson, Thomas Hormenu, Lilian S. Mabundo, Christopher W. DuBose, Ram Jagannathan, Peter L. Whitesell, Annemarie Wentzel, Margrethe F. Horlyck-Romanovsky, Anne E. Sumner

**Affiliations:** 1Section on Ethnicity and Health, Diabetes, Endocrinology, and Obesity Branch, National Institute of Diabetes and Digestive and Kidney Diseases, Bethesda, MD 20892, USA; zoe.waldman@nih.gov (Z.C.W.); blayne.schenk@nih.gov (B.R.S.); gduhuze@ughe.org (M.G.D.K.); arielle.patterson@nih.gov (A.C.P.); thormenu@ucc.edu.gh (T.H.); lilian.mabundo@nih.gov (L.S.M.); christopher.dubose@nih.gov (C.W.D.); annemarie.wentzel@nih.gov (A.W.); margrethehr@brooklyn.cuny.edu (M.F.H.-R.); 2National Institute of Minority Health and Health Disparities, Bethesda, MD 20892, USA; 3Institute of Global Health Equity Research, University of Global Health Equity, Kigali 6955, Rwanda; 4Department of Health, Physical Education, University of Cape Coast, Cape Coast P.O. Box 5007, Ghana; 5Department of Medicine, Emory University School of Medicine, Atlanta, GA 30322, USA; ram.jagannathan@emory.edu; 6Howard University Hospital Sleep Disorders Center, Howard University, 2041 Georgia Ave, NW, Washington, DC 20060, USA; peter.whitesell@howard.edu; 7Department of Health and Nutrition Sciences, Brooklyn College, City University of New York, New York, NY 11210, USA

**Keywords:** African immigrants, perceived stress, sleep quality, socioeconomic status

## Abstract

To identify determinants of daily life stress in Africans in America, 156 African-born Blacks (Age: 40 ± 10 years (mean ± SD), range 22–65 years) who came to the United States as adults (age ≥ 18 years) were asked about stress, sleep, behavior and socioeconomic status. Daily life stress and sleep quality were assessed with the Perceived Stress Scale (PSS) and Pittsburgh Sleep Quality Index (PSQI), respectively. High-stress was defined by the threshold of the upper quartile of population distribution of PSS (≥16) and low-stress as PSS < 16. Poor sleep quality required PSQI > 5. Low income was defined as <40 k yearly. In the high and low-stress groups, PSS were: 21 ± 4 versus 9 ± 4, *p* < 0.001 and PSQI were: 6 ± 3 versus 4 ± 3, *p* < 0.001, respectively. PSS and PSQI were correlated (r = 0.38, *p* < 0.001). The odds of high-stress were higher among those with poor sleep quality (OR 5.11, 95% CI: 2.07, 12.62), low income (OR 5.03, 95% CI: 1.75, 14.47), and no health insurance (OR 3.01, 95% CI: 1.19, 8.56). Overall, in African-born Blacks living in America, daily life stress appears to be linked to poor quality sleep and exacerbated by low income and lack of health insurance.

## 1. Introduction

African-born Blacks who migrate to the United States as adults are faced with the stress of changing countries, cultures, and continents [1,2,3]. Stress results from a combination of factors, some of which need immediate attention such as obtaining housing, food, and employment and some of which are chronic, including the events which precipitated immigration [2,4,5,6].

Between 2000 and 2016, the Black African immigrant population in the United States more than doubled from 574,000 to 1.6 million [1,7]. Furthermore, between 2010 and 2018, the African-born Black population living in the United States increased 52% compared to 12% for the overall foreign-born population [1,7]. Hence, African-born Blacks are one of the fastest growing segments of the immigrant population in the United States [1,7]. Thus, it is important to understand the specific and unique needs of African immigrants [8,9].

Stress both promotes and exacerbates cardiometabolic disease [6]. Populations of African descent experience disproportionately high rates of cardiometabolic disease including hypertension, stroke, diabetes, and heart failure [10]. Recognizing factors that trigger stress in African descent populations is critical to the design of interventions which can promote health while mitigating adverse effects [10].

One aspect of stress is the challenge of coping with daily life, and another is long term stress such as living with the memory and consequences of adverse events which occurred over a lifetime [4,5]. The Perceived Stress Scale (PSS) was designed to assess the perception of daily life stress and be universally applicable (Appendix A) [4,5].

Reasons for African-born Blacks moving to the United States include seeking asylum/refugee status, work, study, family reunification, and the diversity visa program [2]. Independent of why Africans have come to live in the United States, they must adapt daily to activities necessary to become incorporated into American society. Prior to the development of PSS by Cohen et al., assessment tools did not distinguish between negative life events and the stress induced by the “hassles of daily life” such as paying rent, buying food, finding a job, acquiring health insurance, and ensuring child and elder care [4]. In short, not considering the cost of coping with daily life is to underestimate the full impact of stress [4,5].

The INTER-HEART study is a global case-control study of risk factors for myocardial infarction [11]. Initiated in 2000, data from 46 countries across the economic spectrum were accessed and questions about stress were included [11,12]. The stress survey used in the INTER-HEART study has been widely administered including in studies of African immigrants [13,14]. However, the INTER-HEART study does not distinguish between daily life stress and chronic stress. Allowable answers to queries about stress related to work, home and finances are: “never”, “some of the time”, “permanent” or “in the past 12 months” [12,13,14]. Therefore, events or circumstances related to daily life stress are subsumed within longer time intervals. Hence the unique contribution and precipitators of daily life stress cannot be delineated.

Sleep quality and perceived stress are intertwined [15,16]. The Pittsburgh Sleep Quality Index (PSQI) measures sleep quality in the month prior to questionnaire administration [17]. Therefore, both PSQI and PSS were created to understand the impact and consequences of daily life struggles based on a one-month recall period.

To our knowledge PSS and PSQI have not been administered simultaneously in other African immigrant cohorts such as the Afro-Cardiac Study [18], the African Immigrant Health Study [19], Research on Obesity and Diabetes in African Migrants (RODAM) [13,14,20,21], and the Africans in America study [2,22,23,24]. The Afro-Cardiac Study [18] revealed that cardiovascular disease (CVD) risk in West Africans living in the Baltimore-Washington area depends on degree of acculturation and duration of United States residence. The African Immigrant Health Study [19] has illuminated the adverse influence of discrimination on CVD risk in Africans from West and Central Africa. The RODAM [13,14,20,21] study has provided key insights into obesity, exercise, CVD, and diabetes physiology in Ghanaians who have migrated to Amsterdam, Berlin, and London. The Africans in America [2,22,23,24] cohort [2,22,23,24] enrolls African-born Blacks living in metropolitan Washington, DC and previously focused on issues related to both metabolism and chronic stress such as reason and age of immigration. With this report we provide our newest insights from the Africans in America cohort regarding contributors to daily life stress.

To understand daily life stress and sleep quality in African immigrants, PSS and PSQI were administered to 156 African-born Blacks who came to the United States as adults (age ≥ 18 years).

## 2. Materials and Methods

The Africans in America cohort was established to assess both the well-being and cardiometabolic health of African-born Blacks living in the United States [23,25,26,27,28]. The recruitment methods included announcements on the NIH website, flyers, presentations at community events (in-person and virtual) and previous participant referrals. The NIDDK Institutional Review Board approved the protocol (ClinicalTrials.gov Identifier: NCT00001853). Written informed consent was obtained from every participant.

Qualification for enrollment was determined by a telephone interview. Potential participants had to be born in a sub-Saharan African country and be between the ages of 18 and 70 years, currently live in the metropolitan Washington DC area, identify as Black, and report that both parents identified as Black and were also born in a sub-Saharan African country. Persons with a previous diagnosis of diabetes were excluded.

Between 2016 and 2021, 218 consecutively enrolled African-born Blacks participated in two outpatient visits conducted at the NIH Clinical Center, Bethesda, Maryland (Figure 1).

At Visit 1, social and medical histories were taken, and a physical examination, an electrocardiogram and routine blood tests were performed. Thirty-two participants were excluded because they were childhood immigrants, meaning their age at United States entry was <18 years. Childhood arrivals face different educational financial and acculturation pressures than adult arrivals [23]. Therefore, they were not included in the current analyses. The remaining 186 enrollees were adulthood immigrants as they had entered the United States at age ≥ 18 years. Thirty adulthood immigrants did not proceed from Visit 1 to Visit 2. The two most common reasons for not proceeding to Visit 2 were either anemia discovered at Visit 1 or scheduling conflicts. This study reports on the 156 participants who entered the United States as adults and proceeded to Visit 2.

At Visit 2, PSS and PSQI were administered. Weight, height, blood pressure (BP), waist circumference (WC), hemoglobin A1C and hemoglobin electrophoresis were obtained. In addition, oral glucose tolerance tests (OGTT) (Trutol 75, Custom Laboratories) were performed.

### 2.1. Perceived Stress Scale

Perceived stress was measured using the 10-question version of the PSS (Appendix A) [4,5]. Each question begins with “In the past month” and is scored with 0 to 4 points with the maximum score being 40. For diagnosis of high-stress, PSS was converted into a dichotomous variable. In the absence of a universally accepted threshold for high-stress, PSS thresholds are determined on a population basis. While some investigators have chosen to use the threshold at the upper third of PSS in their population distribution [15], we were more rigorous and used the threshold at the upper quartile of PSS in our population (PSS ≥ 16).

### 2.2. Pittsburgh Sleep Quality Index

The PSQI assesses sleep quality over the past month [17]. The PSQI has 19 questions in seven areas, specifically: (1) subjective sleep quality, (2) sleep latency (how long it takes to fall asleep), (3) sleep duration, (4) habitual sleep efficiency (relates total hours of sleep to total hours spent in bed), (5) sleep disturbances (e.g., bad dreams, snoring, bathroom use), (6) use of sleeping medication, and (7) daytime dysfunction (including daytime sleepiness and level of enthusiasm). Each of the 7 components is given a score between 0 and 3. The PSQI score has the potential to range from 0 to 21. The higher the PSQI score, the lower the sleep quality. A PSQI > 5 indicates poor sleep quality because of severe difficulties in two areas or moderate difficulties in three or more areas [17].

### 2.3. Social Variables

Socio-economic variables were dichotomized and defined as education according to yes or no answers for college graduation, health insurance and partnered. Partnered was defined as married or living with significant other or not partnered (never married, separated, divorced or widowed) [3]. Low income was defined as <40 k/year. Gender was defined as male or female.

Behavioral variables included: smoking (yes or no in the last year), alcohol intake (one or more drinks per week vs. less than once per week) and physical activity. Physical activity was determined by the International Physical Activity Questionnaire (IPAQ) categories and dichotomized as sedentary (IPAQ category Low) versus active (IPAQ category Moderate and High combined) [29].

Migration related issues included reason for moving to the United States, age at United States entry, and duration of residence in the United States (<10 years) [2,23,30]. High stress reasons for immigration were seeking work or asylum/refugee status [2,30]. Low stress reasons were coming to the United States to study, family reunification or diversity visa lottery [2,30]. Note, all participants entered the United States at age 18 years or greater. The range for age at United States entry was 18 to 56 years.

### 2.4. Assays

Hemoglobin and hematocrit were measured in EDTA-anticoagulated whole blood using a Sysmex XE-5000. A1C levels were determined by HPLC using the National Glycohemoglobin Standardization Program (NGSP). Hemoglobin electrophoresis was performed with a Helena Zip-Zone electrophoresis instrument (Helena Laboratories, Beaumont, TX, USA).

### 2.5. Statistical Analyses

Unless stated otherwise, data are presented as mean ± SD. For the analyses of the three African regions of origin (Table 1), continuous variables were compared by one-way ANOVA with Bonferroni corrections for multiple comparisons. For comparison of the high and low-stress groups (Table 2), comparisons were by un-paired *t*-tests. Categorical variables in both tables were compared by chi-square and Dunn tests, as appropriate.

High-stress was defined by the upper quartile of the population distribution for PSS. Then we performed a backward stepwise logistic regression to determine the association of 10 a priori social variables with the high-stress group. The 10 social variables were: poor sleep quality (PSQI > 5), income < 40 K, health insurance, life partner status, alcohol intake, education, reason for immigration, sedentary behavior, gender, and duration of United States residence (Table 3). As the prevalence of smoking was only 5%, a meaningful odds ratio could not be calculated, and smoking was not entered into the regression.

Poor sleep quality was defined by Buysse et al. [17] as PSQI > 5. To elucidate the relationship between PSS and total PSQI, Pearson correlation coefficients were calculated. In addition, as the PSQI has seven components, the relative importance of each to PSS as a continuous variable was determined in a multiple linear regression. PSS was the dependent variable and the seven components of the PSQI were the independent variables. For the seven components, collinearity was tested using collinearity diagnostics. No evidence of multicollinearity between covariates was observed for any of the fitted models (variance inflation factor < 1.5 for all independent variables).

*p*-values ≤ 0.05 were considered significant. Data were managed with Research Electronic Data Capture (REDCap) [31]. Analyses were performed with STATA17 (College Station, TX, USA).

## 3. Results

The 156 African-born Black participants (male: 60%; age: 40 ± 10 years, range 22 to 65 years; BMI: 27.6 ± 4.2 kg/m^2^, range 19.3 to 39.5 kg/m^2^) were mainly from countries in West (37%), Central (14%) and East (49%) Africa (Table 1). The four participants from Southern African countries were included in the East African group.

Comparison by African region of origin revealed that BMI was highest in West Africans and income lowest in Central Africans. These differences by BMI and income are also true in sub-Saharan Africa [32]. For the other parameters including demographic and metabolic characteristics, PSS, PSQI scores, health behaviors and migration related factors, there were no significant differences by African region of origin (Table 1) and therefore, the cohort was evaluated independent of African region of origin.

### 3.1. High and Low-Stress Group Comparisons

Neither demographic nor metabolic characteristics differed by stress group, but the high-stress group had higher PSS and PSQI scores than the low-stress group (both *p* < 0.001) (Table 2).

In addition, income was lower and health insurance coverage less frequent in the high-stress group (both *p* < 0.001). Having no life partner was also more common in the high-stress group than the low-stress group (*p* = 0.016), but academic achievement as measured by college graduation rate did not differ by group (*p* = 0.380).

For the three adverse health behaviors studied, smoking was more frequent in the high-stress than the low-stress group (*p* = 0.012) (Table 2). However, the overall prevalence of smoking was only 5%. In contrast the prevalence of both sedentary behavior and alcohol intake > 1 drink/week was 22%, but there was no statistical difference by stress group in alcohol intake or sedentary behavior.

For migration related variables, neither age at United States entry nor reason for immigration to the United States were different in the high-stress versus the low-stress group (Table 2). The total number of years lived in the United States was less in the high-stress than the low-stress group (7 ± 9 vs. 10 ± 10 years, *p* = 0.160) but the difference did not achieve significance. However, the percentage of individuals who lived in the United States < 10 years tended to be higher in the high-stress than the low-stress group (67% vs. 60%, *p* = 0.054).

In a separate analysis the cohort was stratified by United States residence < 10 years or ≥10 years. The percent of individuals with income < 40 k was higher in the group with <10 years residence than ≥10 years (68% vs. 29%, *p* < 0.001).

### 3.2. Odds of Being in the High-Stress Group

By backward stepwise logistic regression, the odds of being in the high-stress group were calculated for 10 social variables (Table 3). The significant odds ratios were: poor sleep quality (PSQI > 5) (OR: 5.11, 95% CI: 2.07, 12.62), income < 40 k (OR: 5.03, 95% CI: 1.75, 14.47), and no health insurance (OR: 3.01, 95% CI: 1.19, 8.56). For having no life partner, the *p*-value approached significance at 0.07, but no odds ratio was calculated because the threshold of <0.05 was not reached. For smoking, the overall prevalence was so low at 5% (Table 2) that no meaningful odds ratio could not be calculated, but among those who smoked, the prevalence of smoking was four times higher in the high-stress group than the low-stress group (13% vs. 3%, *p* = 0.012).

### 3.3. Perceived Stress and Sleep

The Pearson correlation coefficient of PSS and PSQI was highly significant (r = 0.38, *p* < 0.001). Next, we determined which of the seven components of PSQI had the greatest influence on PSS. With PSS as the dependent variable and the seven components of the PSQI as independent variables (adjR^2^ = 32%), multiple linear regression revealed that three of the seven components of the PSQI had significantly influenced PSS. They were sleep disturbance (*p* < 0.001), daytime dysfunction (*p* < 0.001) and subjective sleep quality (*p* = 0.002) (Table 4). In addition, decreased sleep duration tended to increase PSS (*p* = 0.078).

In additional analyses regarding sleep, the participants reported that the number of hours slept in the United States were less than in Africa (*p* < 0.001) and that their self-reported health was worse in the United States than in Africa (*p* < 0.001).

## 4. Discussion

In this first evaluation of both PSS and PSQI in African-born Blacks living in America, we found that the stress of daily life was increased by low income, no health insurance and poor sleep quality. Two factors which are linked to low income and were more common in the high-stress than the low-stress group were shorter duration of United States residence and having no life partner.

### 4.1. Income and Health Insurance

Low income was associated with high-stress (OR: 5.03, 95% CI: 1.75, 14.47). Having sufficient finances to meet immediate needs such as paying for food and housing is a paramount challenge of life in the United States. But, beyond food and housing, there are two other forms of financial pressure which Africans in America enrollees experience: (a) the cost of health insurance; and (b) sending remittances to African relatives.

Independent of race/ethnicity and nativity, lack of health insurance in the United States is a major source of anxiety. The lack of health insurance is often a proxy for low income [33], but in the Africans in America cohort a lack of health insurance, even after adjustment for income, was associated with higher odds of being in the high-stress group (OR: 3.01, 95% CI: 1.19, 8.56). Therefore, for African immigrants, as with others living in the United States, no health insurance and low income are twin measures of economic pressure and daily life stress.

Adding to the economic pressure experienced by immigrants is the frequency with which money is sent back to families living in Africa [34]. Currently, we have little data on the remittance practices of our participants. But historically, the magnitude of the funds sent by immigrants to African-based relatives is so sizable that as much as 15% of the gross domestic product in several sub-Saharan African countries is accounted for by remittances [7]. As remittances are viewed by many immigrants as an essential expense, remittances are deducted from disposable income and less money is available to meet daily needs in the United States.

### 4.2. Duration of United States Residence

Another challenge relative to income pertains to the duration of United States residence. The percent of individuals with income < 40 k was higher in the group with <10 years residence than >10 years (68% vs. 29%, *p* < 0.001). As longer duration of stay in the United States was associated with a higher income, longer duration of residence in the United States could be viewed as beneficial. However, studies which focused on cardiometabolic health have reported that residence in the United States for ≥10 years is associated with more hypertension, diabetes, and heart disease than United States residence < 10 years [2,23,35,36]. Hence, residence in the United States ≥ 10 years has a double edge. Long term residence in the United States is beneficial because of higher income which mitigates daily life stress, but simultaneously it is associated with worse cardiometabolic health.

### 4.3. Life Partner

Having a life partner may represent both social support and less economic pressure, assuming the two people are sharing expenses such as rent. Consistent with this social framework, the lack of a life partner was more frequent in the high-stress than the low-stress group (62% vs. 39%, *p* = 0.016), but an odds ratio was not calculated for having no life partner, because in the logistic regression for high-stress group membership the *P*-value for no life partner did not reach significance (*p* = 0.070) (Table 3).

### 4.4. Sleep

The relationship between sleep quality and stress is known to be bidirectional, which means that each magnifies the adverse effect and consequences of the other [15,16,37]. Correspondingly, we found that PSQI and PSS were significantly correlated (r = 0.38, *p* < 0.001) and the odds of being stressed were increased if PSQI > 5 (OR:5.11, 95% CI:2.07, 12.62).

To gain insight into which of the seven components of the PSQI contributed most to perceived stress in African immigrants, we used the PSS score as the outcome variable and the seven components of PSQI as the independent variables. Three of the seven PSQI components increased the PSS score, specifically sleep disturbances, daytime dysfunction, and poor sleep quality (all *p* < 0.01). In addition, the effect of sleep duration on stress approached significance (*p* = 0.078).

Both long and short sleep duration have been associated with adverse cardiometabolic health [15,37,38,39,40]. The physiologic pathways that associate short sleep duration with adverse cardiometabolic health have been well delineated. Short sleep duration induces changes in insulin resistance, inflammatory factors and appetite related cytokines [39,40]. From the perspective of the Africans in America cohort, participants reported that their nighttime sleep duration was less in the United States than it had been in Africa. They also believed that their health in the United States was worse than it had been in Africa. Based on our current understanding of short sleep and health, we speculate that African immigrants may consider their health to be worse in the United States because they sleep less.

### 4.5. Smoking

In the Africans in America cohort, 5% of participants reported that they were smokers. A similarly low prevalence of smoking has been reported in both RODAM and the Afro-Cardiac cohorts [18,41]. Despite the low overall prevalence of smoking, the prevalence of smoking was four times higher in the high-stress group than the low-stress group (13% vs. 3%, *p* = 0.012). These findings reinforce the concept that smoking paradoxically increases stress even as it is simultaneously used as a coping strategy to decrease stress [5,42,43].

### 4.6. Chronic Life Stress

Precipitants of chronic stress are different from the factors which trigger daily life stress as measured by the PSS [4,5]. In previous work with the Africans in America cohort, we found that chronic stress as measured by allostatic load score (ALS) was significantly higher in those who came to the United States seeking asylum than those who came for family reunification [2]. However, being in the high-stress group as measured by PSS was not affected by reason for immigration, a chronic stressor. In short, the lack of influence of a global life event such as reason for immigration was built into the design of PSS.

Cohen and Williamson state that the reliability of PSS as a predictive tool of stress rapidly declines at four to eight weeks [4]. Therefore, differences in the effect of financial stress, an acute stressor, and reason for immigration, a chronic stressor, on PSS illustrate that PSS is assessing African immigrants as it was designed to.

### 4.7. Education, Gender, Physical Activity and Alcohol Intake

Four factors which did not affect being in the high-stress group were: education, gender, exercise, and alcohol intake. The consensus is that higher education should be associated with higher income [4,5]. Even though African immigrants are highly educated, many Africans received their degrees from universities outside of the United States. Therefore, employment opportunities in the United States may not match educational attainment.

Women in the United States appear to experience more stress than men [4,5,44,45]. However, among African immigrants we found that stress did not differ by gender. Both cultural expectations and immigration patterns may explain this difference. The majority of immigrants from Africa to the United States as well as in the Africans in America cohort are men [46].

In the Africans in America cohort, sedentary behavior occurred in 22% of the cohort and did not differ by stress group. Importantly, the prevalence of sedentary behavior is 21% in sub-Saharan Africa and 23% in Ghanaian migrants to Europe [20,47]. The underutilization of physical exercise as a stress management tool may be explained by differences in cultural preferences and social norms. In the United States, there is an increasing cultural awareness of the value of physical activity. In contrast, Afrifa-Anane et al. reported that, in the RODAM study, physical inactivity in Ghanaian migrants to Europe was inversely correlated with social standing [20].

Alcohol intake did not vary by stress group. However, alcohol intake was evaluated as a dichotomous variable, specifically any alcohol intake versus no alcohol intake. Notably, Cohen et al. found that, when drinking was dichotomized between no drinking and drinking, no relationship between alcohol intake and PSS was detected [4,5]. However, Cohen et al. also found that among people who drank, higher intake was associated with higher stress [4,5]. In the Africans in America cohort, only 22% of participants reported any alcohol intake. Due to the small number of people who reported any alcohol intake, it was not possible for us to assess the relationship between quantity of alcohol consumed and stress.

### 4.8. Strengths and Limitations

From two perspectives, data from the Africans in America cohort can be used to support the use of PSS and PSQI in African immigrants. First, the PSS was designed to detect daily life stress and not stress related to global life events. In the Africans in America cohort, with PSS as the outcome variable, low income, a daily life stressor, significantly affected PSS, but reason for immigration, a chronic stressor did not. Second, and as found in other populations, PSS and PSQI were highly correlated [15,37,48,49].

Limitations of our study include the cross-sectional design, the use of a convenience sample and the relatively small sample size. However, our recruitment area includes not only metropolitan Washington, DC but also Montgomery County, MD and Prince George’s County, MD. All three of the areas are cited among the places in the United States with the highest concentrations of African-born Blacks [7]. Furthermore, the sample appears to be representative of African immigrants because the prevalence of cigarette smoking, sedentary behavior and diabetes were similar to data from other African immigrant cohorts as well as statistics from national databases [18,20,41,50]. In addition, and consistent with known immigration patterns, as most of the participants were male, immigration was more common from countries in West and Eastern Africa than central Africa and BMI was highest in West Africans [7,32]. Moreover, the frequency of sickle cell trait and hemoglobin C trait in the Africans in America cohort mirror known differences in prevalence by African region of origin [51,52].

Another weakness which can be addressed in future investigations is that neither perceived discrimination nor resilience were assessed in this investigation. This is important because discrimination occurs in daily life and profoundly influences both daily life stress and sleep quality [53]. Further, resilience can be viewed as an antidote to stress and the Connor-Davidson Resilience scale provides a 30-day measure of resilience [54].

## 5. Conclusions

Both daily life stress and sleep quality as assessed by PSS and PSQI provide important insight into the experience of African immigrants. Economic pressure and sleep quality are two factors which adversely affect daily stress and are potentially modifiable. Therefore, policy makers should consider providing services to immigrants which include financial counseling and information on sleep hygiene. Primary care providers along with clinical researchers should work to optimize the health and well-being of African immigrants by collecting data to assess daily life stress and sleep quality as well as perceived discrimination, resilience, and chronic stress.

## Figures and Tables

**Figure 1 ijerph-19-02562-f001:**
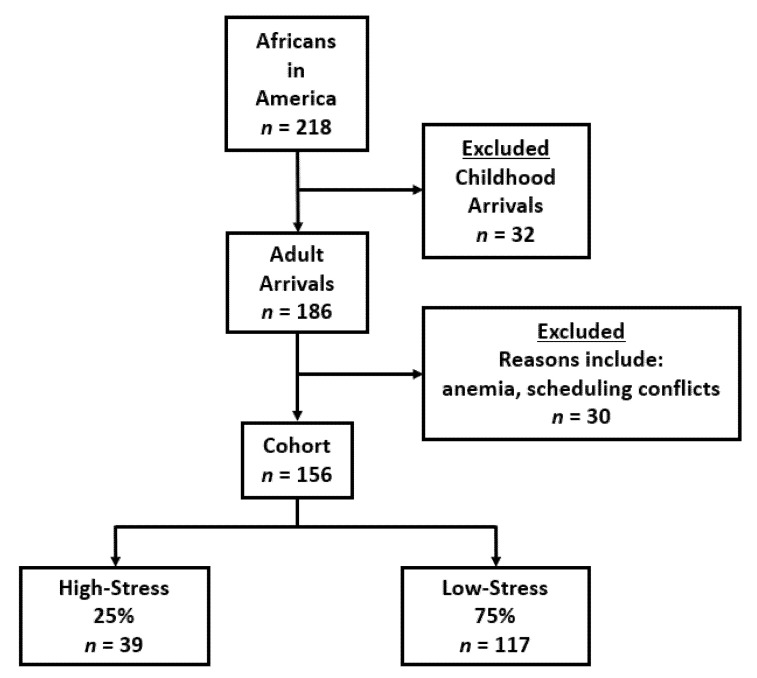
Flow Chart for Enrollment.

**Table 1 ijerph-19-02562-t001:** Demographic, Metabolic and Social Characteristics by African Region of Origin.

Parameters ^1^	Cohort 100% (*n* = 156)	West 37% (*n* = 58)	Central 14% (*n* = 22)	East ^2^ 49% (*n* = 76)	*p*-Value ^2,3^
Demographics					
Sex (% male)	60%	73%	54%	64%	0.224
Age (years)	40 ± 10	40 ± 11	40 ± 11	40 ± 10	0.941
Metabolic characteristics					
Hemoglobin (g/dL)	13.9 ± 1.5	13.8 ± 1.5	13.9 ± 1.5	13.9 ± 1.5	0.964
Sickle cell trait or HbC trait (%)	14%	19%	18%	7%	0.081
Body mass index (kg/m^2^)	27.6 ± 4.2	28.7 ± 4.2	26.9 ± 3.9 a *	26.9 ± 4.2	0.042
Waist circumference (cm)	91 ± 12	93 ± 11	88 ± 10	91 ± 12	0.231
Systolic blood pressure (mmHg)	117 ± 12	118 ± 11	119 ± 10	118 ± 10	0.630
Diastolic blood pressure (mmHg)	71 ± 9	71 ± 9	71 ± 10	71 ± 9	0.869
Fasting glucose (mg/dL)	94 ± 16	96 ± 23	95 ± 8	92 ± 10	0.455
A1C (%)	5.4 ± 0.7	5.5 ± 1.0	5.5 ± 0.5	5.4 ± 0.4	0.609
Diabetes (%)	8%	10%	5%	7%	0.602
Scores					
Perceived stress score (PSS)	12 ± 7	11 ± 6	12 ± 7	12 ± 7	0.808
Pittsburgh sleep quality score (PSQI)	4.8 ± 2.9	4.9 ± 3.0	5.1 ± 3.3	4.6 ± 2.8	0.726
Socioeconomic					
Income < 40 k	54%	57%	77% a *	53%	0.046
No health insurance	39%	40%	46%	37%	0.762
No partner (%)	45%	48%	50%	41%	0.601
Education (no college degree)	23%	14%	27%	29%	0.105
Adverse health behaviors					
Smoking	5%	5%	5%	5%	0.991
Alcohol ≥ 1 drink/week	22%	24%	18%	22%	0.850
Sedentary	22%	24%	18%	22%	0.850
Migration factors					
Age at United States entry (years)	31 ± 9	30 ± 9	32 ± 9	32 ± 9	0.542
High risk immigration reason ^4^	24%	19%	41%	24%	0.122
Years in United States (years)	9 ± 10	10 ± 11	9 ± 8	9 ± 8	0.648
United States residence < 10 years	64%	60%	59%	68%	0.546

^1^ Data presented as mean ± SD or percent. ^2^ For continuous variables: comparisons were by one-way ANOVA with Bonferroni corrections for multiple comparisons; for categorical variables: comparisons were by Chi-square. ^3^ a: West vs. Central, b: West vs. East, c: Central vs. East, * ≤0.05, ** ≤0.01, *** ≤0.001. ^4^ High risk immigration reason included seeking Work or Asylum/Refugee Status.

**Table 2 ijerph-19-02562-t002:** Characteristics According to Stress Group.

Variable ^1^	Total 100% *n* = 156	High-Stress ^2^ 25% *n* = 39	Low-Stress 75% *n* = 117	*p*-Value ^3^
Demographics				
Sex (% male)	60%	56%	62%	0.571
Age (years)	40 ± 10	38 ± 10	41 ± 10	0.242
Metabolic characteristics				
Hemoglobin (g/dL)	13.9 ± 1.5	13.7 ± 1.7	13.9 ± 1.4	0.386
Body mass index (kg/m^2^)	27.6 ± 4.2	27.7 ± 4.0	27.5 ± 4.0	0.863
Waist circumference (cm)	91 ± 12	91 ± 12	91 ± 12	0.942
Systolic blood pressure (mmHg)	117 ± 12	117 ± 13	117 ± 12	0.852
Diastolic blood pressure (mmHg)	71 ± 9	71 ± 9	71 ± 9	0.848
Fasting glucose (mg/dL)	94 ± 16	92 ± 9	95 ± 18	0.350
A1C (%)	5.4 ± 0.7	5.3 ± 0.5	5.5 ± 0.8	0.260
Diabetes	8%	8%	8%	0.999
Scores				
Perceived stress scale	12 ± 7	21 ± 4	9 ± 4	<0.001
Pittsburgh sleep quality index (PSQI)	4.8 ± 2.9	6.2 ± 2.9	4.3 ± 2.8	<0.001
Poor sleep quality (PSQI > 5) (%)	39%	64%	30%	<0.001
Socioeconomic				
Income < 40 k (%)	54%	85%	44%	<0.001
No health insurance (%)	39%	67%	30%	<0.001
No partner (%)	45%	62%	39%	0.016
Education (no college degree)	23%	28%	21%	0.380
Adverse health behaviors				
Smoking	5%	13%	3%	0.012
Alcohol ≥ 1 drink/wk	22%	18%	24%	0.438
Sedentary	22%	21%	23%	0.740
Migration factors				
Age at United States entry (years)	31 ± 9	31 ± 9	31 ± 9	0.779
High Risk Immigration Reason ^4^	24%	28%	23%	0.518
Years in United States (years)	9 ± 10	7 ± 9	10 ± 10	0.160
United States residence < 10 years	64%	67%	60%	0.054

^1^ Data presented as mean ± SD or percentages. ^2^ Upper quartile of PSS for the population distribution (PSS ≥ 16). ^3^ For continuous variables: comparisons were by unpaired *t*-test; for categorical variables: comparisons were by Chi-square. ^4^ High Risk Immigration reasons included seeking Work or Asylum/Refugee Status.

**Table 3 ijerph-19-02562-t003:** Backward stepwise logistic regression to determine odds of being in the high-stress group.

Variable	Odds Ratio (95% CI)	*p*-Value
Poor sleep quality (PSQI > 5)	5.11 (2.07, 12.62)	<0.001
Income < 40 k	5.03 (1.75, 14.47)	0.002
No health insurance	3.01 (1.19, 8.56)	0.019
Single (no life partner)	X	0.070
Alcohol intake (≥1 drink/wk)	X	0.260
education ^1^	X	0.424
High risk immigration ^2^	X	0.669
Sedentary	X	0.743
Gender (male)	X	0.856
United States Residence < 10 years	X	0.890

^1^ No college degree ^2^ High Risk Immigration reasons were Seeking Work or Asylum/Refugee Status; Low risk were study, family reunification and diversity visa program.

**Table 4 ijerph-19-02562-t004:** Multiple Linear Regression to Determine the Influence of each PSQI component on PSS.

PSQI Components	β-Coefficient (95% CI)	*p*-Value
Sleep Disturbance	3.51 (1.87, 5.16)	<0.001
Daytime Dysfunction	2.27 (1.09, 3.45)	<0.001
Subjective Sleep Quality	2.07 (0.75, 3.39)	0.002
Sleep Duration	−0.86 (−1.08, 0.10)	0.078
Sleep Medicine	−1.58 (−3.40, 0.26)	0.091
Sleep Latency	0.71 (−0.22, 1.65)	0.122
Habitual Sleep Efficiency	−0.44 (−1.60, 0.74)	0.463

## Data Availability

Data available upon reasonable request.

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
