# Peer review of "Sleep and Economic Status Are Linked to Daily Life Stress in African-Born Blacks Living in America"

_ijerph, 2022, doi:10.3390/ijerph19052562_

Round 1

Reviewer 1 Report

A number of life stressors, particularly the economic situation of African-born Blacks residing in America, were addressed in the work.
For a variety of unavoidable circumstances, the article became centered on just one sort of migrant from one continent, who belonged to a very specific age range, which constrained the scope of the research and affected it in a number of important ways. In addition, for some reason, the data on immigration status cited in the manuscript is extremely old, having been collected in 2013. (lines 39). For reference, I hope it's easily accessible so that readers can acquire current data. I would love to have some further explanation on the selection of certain color migrants rather than a mix population of migrants from various continents and cultures included in the manuscript. Also, I would urge authors to think about presenting some future considerations for policymakers and clinicians to keep in mind in the future when reading this work.

Author Response

Reviewer 1

We thank the Reviewer. The manuscript and particularly the introduction has been significantly revised and improved by responding to the Reviewer’s concerns.

Citing the most recent online publications of the Pew Research Center on Key Facts about African immigration from January 24, 2018 and the Migration Policy Institute published in 2019, we now state that the Black African immigrant population in the United States has more than doubled from 2000 to 2016 and grown from 574,000 to 1.6 million. Between 2010 and 2018, the African immigrant population in the United States increased by 52% compared to 12% for the overall immigrant population. Therefore African-born Blacks are among the fasting growing segments of the immigrant population in the United States (pg. 2, para 2).

The focus is on African immigrants because of the increasing appreciation that to optimize their care and well-being, the specific needs and determinants of stress in this population needs delineation. In short, there is growing awareness of the diversity within African descent populations (Commodore-Mensah et al. ref 8 and Utumatwishima et al ref 9).  Africans coming to the United States may be confronted with abrupt changes in economic and social status and family structure in ways that are fundamentally different from people of African descent coming to the United States from Caribbean, Central and South America.

We now provide more information about the age range of our participants. The age range for enrollment per our IRB application is 18 to 70 years (pg.3, para 4, ln 2-3) The age range of people who actually enrolled was 22 to 65 years (pg 6, para 4 ln 1) This age ranges covers young adults, middle age-adults and middle age-adults approaching retirement or already retired.

Our recommendations in the final paragraph for policy makers is that counseling on financial matters and sleep hygiene should be built in services for immigrants. In addition, primary care providers and clinical investigators should broaden our understanding of the African immigrant experience by combining data on perceived daily stress and sleep with data on perceived discrimination, resilience, and chronic stress. These recommendations are now included in the final paragraph of the manuscript (pg. 14, para 3 ln 4-8).

Reviewer 2 Report

The authors address an important question, however the current paper has a fundamental methodological flaw. Number of subjects is extremely low to address the question and there is not a control group. Also, choice of cut off for both PSS and PSQI should be explained (literature? (Ref?),  statistical distribution of their own  sample ?

Author Response

Reviewer 2

We thank Reviewer 2 and appreciate the concerns raised. African descent populations are profoundly diverse. There is no control group because this study is an exploration of the factors that lead to stress and poor sleep quality specifically in African immigrants.  For example, Africans due to culture and the reasons which precipitated their immigration may be confronted with abrupt changes in economic and social status and family structure in ways which are fundamentally different from Black immigrants from the Caribbean or Central or South America. In addition, the lived experience of African Americans and African immigrants are profoundly different. Commodore-Mensah et al. (ref 8) and Utumatwishima et al. (ref 9) call for the need to invest resources to understand the specific and unique aspects of different African descent populations and to be sensitive to the need to disaggregate different groups to better understand each.

Sample size of a study of this type is always a concern but the study was large enough to find the expected inverse relationship between PSS and PSQI (Abstract and pg10, para 1, ln 1-2). This is presumptive evidence that our study was sufficiently large to promote the use of PSS and PSQI in future investigations of the challenges African immigrants face. This was a main goal of our investigation as this is the first study to apply to PSS and PSQI in an African immigrant cohort (pg. 3, para 1, ln 1-4).

The cut-off for PSQI of greater than 5 for poor sleep was established by Buysse et al. (ref 17) in the article which established the PSQI as an important tool to measure sleep quality. The title of article is: The Pittsburgh Sleep Quality Index: A New Instrument for Psychiatric Practice and Research. They found that a PSQI scores >5 distinguished good sleepers (n=52) from poor sleepers (n=54).  In addition, they reported that PSQI>5 indicates poor sleep quality because of severe difficulties in two areas or moderate difficulties in three or more of the 7 components of sleep. We agree that this was not sufficiently explained in the original manuscript. Please see p. 5, para 1, ln 8=9

In the absence of a universally accepted threshold for high-stress, PSS thresholds are determined on a population basis. While some investigators have chosen to use the threshold at the upper third of PSS in their population distribution (ref 15), we decided to be more rigorous and use the threshold at the upper quartile of PSS in our population (PSS≥16). This is now better described in pg. 4, para 3, ln 3-8).

Reviewer 3 Report

Review of the article entitled “Sleep and Economic Status are Linked to Daily Life Stress in African-born Blacks living in America”

In this manuscript the authors evaluating the link between sleep quality and economic status to stress in African-born Blacks in America. Several studies demonstrated that racial/ethnic minorities are more likely to experience, poor sleep quality, shorter sleep durations, irregular sleep timing etc. in comparison to Whites. Racial/ethnic groups have significant heterogeneity, there are limited studies focused on the within-group analyses. These sleep health disparities are a significant public health problem that warrants special attention.
The manuscript is quite interesting and very well written, the bibliography is extensive and up to date. I highly recommend to publish this article in the IJERPH. 
However, I have some suggestions for the authors:  
1. Table 1 –I would suggest to remove from the manuscript and add as a supplemental file.
2. Table 3 the Mean and SD should have the same number of decimal points.
3. line-228-229 the authors report “tended to be less” but p value shows no tendency.

Author Response

We thank Reviewer 3 for the support.

However, I have some suggestions for the authors:  
1. Table 1 –I would suggest to remove from the manuscript and add as a supplemental file.

We agree. Furthermore, this suggestion mirrors the request of the Editor to have Table 1 removed from the main document. We have now created a supplemental file for this table.

  1. Table 3 the Mean and SD should have the same number of decimal points.

Table 3 is now Table 2. The mean and SD have the same number of decimal places for all entries in all tables. Thank you!

3. line-228-229 the authors report “tended to be less” but p value shows no tendency.

The P-value is 0.160. The definition of “tended to be less” is vague. Therefore, a formal search on the phrase was undertaken and revealed that most often it is <0.100 or even up to <0.130.  So, we agree that P=0.160 is not sufficiently low to be identified as “tended to be less”. The phrase was deleted from the sentence.  

Round 2

Reviewer 2 Report

The authours have not addressed reviewer's concerns. The number of subjects is extremely low and there is not a control group.

Author Response

We regret that the Reviewer was not satisfied with our response to two issues, (1) our sole focus on African immigrants to the United States and (2) the sample size.

  1. About our sole focus on African immigrants to the United States: Our manuscript was submitted in response to a call for papers to be considered for publication in a Special Volume entitled: The Health of African Migrants: The Burden, Determinants, and Solutions on African Migrant Health.

We suggest that our concentration on African migrants is responsive to the topic of the special volume. Whether other groups of Black Immigrants or even African Americans living in the United States respond to stress, low income or poor sleep quality in ways or degree that are similar to or different from African immigrants was beyond our scope.

  1. About the Size of the cohort. Our sample size was large enough:
    • (a) to be representative of known African immigration patterns regarding gender distribution and African region of origin (ref 1, 7).
    • (b) to be similar to other African immigrant cohorts relative to cigarette smoking (ref 18), sedentary behavior (ref 20), and diabetes prevalence (ref 42 and 51).
    • (c) to identify that both acute and chronic stress have different precipitants  and consequences (ref 4 and 5).
    • (d) to find an inverse relationship between acute stress measured by PSS and sleep quality by PSQI, and thereby validate that these questionnaires are effective in African-born Blacks living in the United States (ref 15 and 16).

We respectfully ask Reviewer 2 to accept our focus on African immigrants and our reasoning about our sample size. We suggest that responses labeled (c  ) and (d) move forward our understanding of African immigrant health in the United States and can be used to justify more and larger studies using PSS and PSQI in African migrants, even though PSS and PSQI were not developed in this population.